# Probing Mechanisms and Therapeutic Potential of γ-Secretase in Alzheimer’s Disease

**DOI:** 10.3390/molecules26020388

**Published:** 2021-01-13

**Authors:** Michael S. Wolfe

**Affiliations:** Department of Medicinal Chemistry, University of Kansas, 1567 Irving Hill Road, GLH-2115, Lawrence, KS 66045, USA; mswolfe@ku.edu

**Keywords:** protease, amyloid, Alzheimer’s disease, inhibitors, modulators

## Abstract

The membrane-embedded γ-secretase complex carries out hydrolysis within the lipid bilayer in proteolyzing nearly 150 different membrane protein substrates. Among these substrates, the amyloid precursor protein (APP) has been the most studied, as generation of aggregation-prone amyloid β-protein (Aβ) is a defining feature of Alzheimer’s disease (AD). Mutations in APP and in presenilin, the catalytic component of γ-secretase, cause familial AD, strong evidence for a pathogenic role of Aβ. Substrate-based chemical probes—synthetic peptides and peptidomimetics—have been critical to unraveling the complexity of γ-secretase, and small drug-like inhibitors and modulators of γ-secretase activity have been essential for exploring the potential of the protease as a therapeutic target for Alzheimer’s disease. Such chemical probes and therapeutic prototypes will be reviewed here, with concluding commentary on the future directions in the study of this biologically important protease complex and the translation of basic findings into therapeutics.

## 1. Introduction

Alzheimer’s disease (AD), the most common form of dementia, is a devastating neurodegenerative disorder that affects perhaps 30 million people worldwide, with demographic projections suggesting this will increase substantially in the coming decades [1]. Cerebral neurodegeneration typically takes place first in the hippocampus, a region below the neocortex that is critical for consolidating long-term memories. Neuronal loss spreads to other cortical areas, leading to progressive cognitive decline. By the end stages of the disease, patients lose cognitive function to the point of requiring constant care, often institutionalized. Although a number of risk factors have been associated with AD, disease onset correlates best with age, and the large majority of cases occur in the elderly. Among people over age 85, over a third are afflicted.

Two types of protein deposits are found in the AD brain: amyloid plaques and neurofibrillary tangles [2]. The former are extraneuronal and primarily composed of the 4 kDa amyloid β-peptide (Aβ), whereas the latter are intraneuronal filaments of the normally microtubule-associated protein tau. Neuroinflammation is a third pathological feature of AD, in which microglia—phagocytic brain immune cells that release cytokines—become overactivated [3]. The role of each of these features in AD etiology and pathogenesis are not well understood. However, Aβ aggregation—in the form of oligomers, protofibrils, fibrils, and plaques—is generally observed as the earliest pathology, followed by tau tangle formation and neurodegeneration [4]. For this reason and those mentioned in the next section, pathological Aβ is widely considered the initiator of AD, triggering downstream tau pathology and neuroinflammation, and Aβ has been the primary target for the development of AD therapeutics for over 25 years [5].

## 2. Familial AD and Genetics

As mentioned above, the “amyloid hypothesis” of AD pathogenesis has reigned for decades, and AD drug development has largely focused on inhibiting Aβ production, blocking Aβ aggregation, or facilitating Aβ clearance from the brain [6]. The primary basis for this dogma is the discovery in the 1990s of dominant genetic mutations associated with early-onset AD [7,8,9,10]. This familial AD (FAD) has a disease onset before age 60 and can occur even before age 30. Other than the monogenetic cause and mid-life onset, FAD is closely similar to the sporadic AD of old age with respect to pathology, presentation, and progression. The most parsimonious explanation is that similar molecular and cellular events are involved in the pathogenesis and progression of both forms of the disease.

The first genetic mutations associated with FAD were in the amyloid precursor protein (APP) [7]. This gene encodes a single-pass membrane protein that is initially cleaved by β-secretase, a membrane-tethered aspartyl protease in the pepsin family, to release the large APP ectodomain [11] (Figure 1). The remnant C-terminal fragment (APP CTF-β) is then proteolyzed within its transmembrane domain (TMD) by γ-secretase to produce Aβ, which is then secreted from the cell [12]. Most Aβ is 40 residues in length (Aβ40), but a small portion is the much more aggregation-prone 42-residue form (Aβ42). Although Aβ42 is a minor Aβ variant produced through APP CTF-β processing by γ-secretase, it is the major form deposited in the characteristic cerebral plaques of AD [13].

The 27 known APP mutations associated with FAD [14], each devastating different families, are all missense mutations in and around the small Aβ region of the large APP. These mutations either alter Aβ production or increase the aggregation tendency of the peptide [15]. A double mutation just outside the *N*-terminus of the Aβ region in APP increases proteolysis by β-secretase, leading to elevated APP CTF-β and therefore elevated Aβ overall. Mutations in the TMD near γ-secretase cleavage sites elevate Aβ42/Aβ40, and mutations within the Aβ region itself make the peptide more prone to aggregation.

FAD mutations were then discovered in presenilin-1 (PSEN1) [8] and presenilin-2 (PSEN2) [9], genes encoding multi-pass membrane proteins that at the time had no known function. These missense mutations—now with over 200 known [14], all but a dozen or so in PSEN1—are located throughout the sequence of the protein but mostly within its nine TMDs [16,17]. Presenilin FAD mutations were soon found to increase Aβ42/Aβ40 [18,19,20], further strengthening the idea that this ratio is critical to pathogenesis. Moreover, these findings indicated that presenilins can modulate γ-secretase cleavage of APP substrate, as the FAD mutations altered the preference for cleavage sites by the protease. Soon after came the observation that knockout of PSEN1 dramatically reduced Aβ production at the level of γ-secretase [21], with the remaining Aβ production attributed to PSEN2 (later verified [22,23]). Thus, presenilins are required for γ-secretase processing of APP CTF-β to Aβ peptides.

## 3. Presenilin and the γ-Secretase Complex

Meanwhile, the design of substrate-based peptidomimetic inhibitors suggested that γ-secretase is an aspartyl protease [24,25,26]. Peptide analogs, based on the γ-secretase cleavage site in the APP TMD leading to Aβ production and containing difluoroketone or difluoroalcohol moieties—mimetics of the transition state of aspartyl protease catalysis—were effective inhibitors of γ-secretase activity in cell-based assays. Given the requirement of presenilin for γ-secretase activity, the site of proteolysis of APP within its TMD, the multi-TMD nature of presenilin, and the evidence that γ-secretase is an aspartyl protease, the possibility was raised that presenilin could be a novel membrane-embedded protease. Indeed, two conserved TMD aspartates were found in presenilins, and both aspartates were required for γ-secretase activity [27]. Subsequent reports that affinity-labeling reagents based on the transition-state analog inhibitors of γ-secretase bound directly to presenilin cemented the idea that presenilin is an unprecedented aspartyl protease with its active site located within the lipid bilayer [28,29].

Although presenilins appeared to be unusual aspartyl proteases, it was clear that they did not have this activity on their own. Presenilins themselves undergo proteolysis within the large loop between TMD6 and TMD7 to form an N-terminal fragment (NTF) and C-terminal fragment (CTF) [30] (Figure 2). The formation of PSEN NTF and CTF is gated by limiting cellular factors [31], and these two presenilin subunits assemble into a larger complex [32,33]. Biochemical analysis and genetic screening ultimately identified three other components of what became known as the γ-secretase complex [34,35,36]. These three components, membrane proteins nicastrin, Aph-1, and Pen-2, assemble with presenilin, activating an autoproteolytic function of presenilin to form PSEN NTF and CTF. [The two essential TMD aspartates of presenilins are also required for PSEN NTF/CTF formation [27].] This assembly with cleaved presenilin is the active form of γ-secretase. Indeed, the transition-state analog affinity labeling reagents that tag presenilins specifically bound to PSEN NTF and CTF [28,29], suggesting that the active site of the protease resides at the interface between these two presenilin subunits. This idea is consistent with the observation that one of the essential aspartates is in TMD6 in the PSEN NTF, and the other is in TMD7 in the PSEN CTF.

Soon after the discovery of presenilin as the catalytic component of γ-secretase, analysis of the other proteolytic product of γ-secretase cleavage of APP (AICD), revealed that the APP TMD was proteolyzed at two different sites [37,38,39,40,41]. Cleavage at the second (ε) site releases AICD products composed of residues 49–99 or 50–99 of the 99-residue APP CTF-β substrate for γ-secretase (Figure 3). With secreted Aβ peptides ranging from 38–43 residues (Aβ38-Aβ43), this left 5 to 11 APP TMD residues unaccounted for. Subsequent discovery of Aβ45, Aβ46, Aβ48, and Aβ49, but no N-terminally extended AICD peptides, led to the hypothesis that ε proteolysis occurs first [42,43,44].

The generated Aβ48 and Aβ49 (counterparts of AICD49-99 and AICD50-99, respectively) were postulated to undergo tripeptide trimming along two pathways: Aβ49→Aβ46→Aβ43→Aβ40 and Aβ48→Aβ45→Aβ42→Aβ38 (this last cleavage step generating a tetrapeptide coproduct). Mass spectrometric analysis of the small peptide products supported this notion [45], as did the finding that synthetic Aβ49 is primarily processed to Aβ40 and Aβ48 is primarily trimmed to Aβ42 by purified γ-secretase [46]. Kinetic analysis of trimming of synthetic Aβ48 and Aβ49 by five different FAD-mutant γ-secretase complexes revealed that all five were dramatically deficient in this carboxypeptidase trimming activity [46].

Presenilin and the γ-secretase complex have many more substrates besides APP [47]. Indeed, so many substrates have been identified that the γ-secretase complex has been called the proteasome of the membrane [48], implying that one of its major functions is to clear out membrane protein stubs that remain after ectodomain release by sheddases. While membrane protein clearance may be an important function of γ-secretase, the protease also plays essential roles in certain cell signaling pathways. The most important of these is signaling from the Notch family of receptors [49]. Notch receptors are single-pass membrane proteins like APP, and proteolytic processing of Notch, triggered by interaction with cognate ligands on neighboring cells, leads to release of its Notch intracellular domain (NICD) (Figure 4). The NICD translocates to the nucleus and interacts with specific transcription factors that control the expression of genes involved in cell differentiation. These signaling pathways, particularly from Notch1 receptors, are essential to proper development in all multi-cellular animals. Knockout of presenilin genes in mice is lethal and leads to phenotypes that are virtually identical with those observed upon knockout of the Notch1 gene [50,51], findings that, as explained later, have major implications for the potential of γ-secretase inhibitors as AD therapeutics.

## 4. Chemical Probes

Small-molecule substrate-based peptidomimetics have been valuable chemical probes in elucidating γ-secretase biochemistry and biology. As mentioned earlier, transition-state analog inhibitors (TSAs; see examples in Figure 5) suggested that γ-secretase is an aspartyl protease [24,25,26], leading to the identification of two conserved TMD aspartates in presenilin essential to γ-secretase activity [27], and TSA affinity probes labeled PSEN NTF and CTF [28,29]. Panels of systematically varied TSAs also helped characterize the nature of the active site, particularly the pockets that accommodate substrate amino acid side chains [25,52,53,54]. In protease terminology, these substrate residues are P1, P2, P3 and so on moving in the *N*-terminal direction from the scissile amide bond and P1′, P2′, P3′ etc. moving in the C-terminal direction. Corresponding pockets on the protease are termed S1, S2, S3 etc. and S1′, S2′, S3′ etc. TSA peptidomimetic probes for γ-secretase suggested (1) relatively loose sequence specificity of the protease, consistent with its nearly 150 other TMD substrates besides APP [55]; (2) pockets S2, S1, S1′ and S3′ are relatively large, accommodating the bulky aromatic phenylalanine side chain, while the S2′ pocket is relatively shallow and does not tolerate phenylalanine; and (3) the protease has three pockets S1′, S2′, and S3′ but apparently no S4′ pocket. This last point has implications for how γ-secretase processes APP substrate: As described earlier, it is now clear that the enzyme carries out multiple proteolytic events in the TMD, trimming down initially formed long Aβ peptides of 48 or 49 residues, generally in intervals of three amino acids, to shorter secreted variants such as Aβ40 and Aβ42. The three pockets S1′, S2′, and S3′ apparently dictate tripeptide carboxypeptidase trimming of long to short Aβ. Moreover, the intolerance of Phe in the S2′ pocket was confirmed by systematic Phe mutagenesis within the APP TMD substrate: In every case where Phe was in the P2′ position relative to a specific cleavage site, proteolysis of that site by γ-secretase was blocked [56].

Immobilization of a TSA inhibitor allowed isolation [57] and ultimately purification [58] of the γ-secretase complex, but this affinity purification approach also revealed something important about how a membrane-embedded protease recognizes substrates. TSA affinity purification of γ-secretase from solubilized cell membranes resulted in co-purification of APP substrate [57]. This was initially surprising, as it was not immediately obvious how substrate could be bound to the enzyme when the active site was occupied by immobilized TSA. However, this makes sense considering that both enzyme active site and substrate cleavage site reside in the membrane. The active site contains two catalytic aspartates that activate a water molecule for hydrolysis of an amide bond. The hydrophilic aspartates and water should be inside the presenilin protein, sequestered from the hydrophobic environment of the lipid bilayer. The substrate is also embedded in the membrane and can only diffuse in two dimensions. To gain access to the internal active site, substrate TMD must first dock on the outer surface of the protease complex. Apparently, during the TSA affinity purification, enzyme was isolated with substrate bound to this initial substrate docking exosite. With the immobilized TSA occupying the active site, substrate remains bound to the external docking site and cannot enter the internal active site for catalytic conversion.

Other substrate-based peptidomimetics were designed to target this docking exosite. With the reasoning that the initial contact with this docking site would involve a classical α-helical conformation of the substrate TMD, peptides based on the APP TMD were synthesized, incorporating the helix-inducing α-aminoisobutyric acid (Aib; α-methylalanine) [59]. In the design, Aib residues were spaced apart every 3–4 residues, so that the Aib residues would lie along one face of the helix, with APP residues arrayed along the rest of the helix and available for binding to the docking exosite (Figure 6). A set of APP-based peptides was synthesized with the placement of the Aib residues staggered to present different faces of the TMD to the protease complex. Additionally, 10-residue l-peptides were identified as low micromolar inhibitors of Aβ production at the γ-secretase level in human cells stably expressing APP. Surprisingly, the corresponding d-peptides, synthesized as controls, were more potent that their mirror-image counterparts. Inverting two internal stereocenters, however, disrupted helicity and dramatically decreased inhibitory potency. Phenylalanine scanning through the ten-residue helical peptide inhibitors (HPIs) led to identification of l- and d-peptides with IC50 values in the mid-nanomolar range. Exploring l- and d-peptides up to 16 residues in length led to discovery of a 13-residue D-HPI with subnanomolar potency [60].

The increased inhibitory potencies of HPIs with phenylalanine in specific positions encouraged the replacement of this residue with the photoactivatable 4-benzoyl-phenylalanine and installation of a linker-biotin on the N-terminus to provide affinity labeling reagents [61]. Such modifications of the potent 10- and 13-residue D-HPIs led to some loss of inhibitory activity but still with low-to-mid-nanomolar potencies. Interestingly, photoactivation of these modified D-HPIs in the presence of HeLa cell lysates resulted in labeling of PSEN1 NTF and CTF. As TSA inhibitors also bind PSEN1 NTF and CTF, competition experiments were run and showed that TSA inhibitor did not compete with the 10-residue D-HPI photoprobe, and the 10-residue D-HPI parent compound did not compete with the TSA photoprobe. This demonstrated that the HPI and TSA compounds, while they both interact with PSEN1 NTF and CTF, bind to distinct sites. This is consistent with the active site residing inside PSEN1 and the docking site residing outside PSEN1, but with both sites at the NTF/CTF interface. Thus, substrate TMD apparently docks and then moves in whole or in part between the NTF and CTF subunits of PSEN1 to access the internal active site. In contrast to the 10-residue D-HPI, the 13-residue D-HPI could compete with the TSA protoprobe, and TSA inhibitor could compete with the 13-residue D-HPI photoprobe. The difference in abilities of the 10- and 13-residue D-HPIs to compete with TSAs suggests that the docking and active sites are proximal, within three amino acids of each other.

## 5. Structural Probes

More recently, HPI and TSA inhibitors have been combined to create substrate-based structural probes for the γ-secretase complex [62]. The structure of the γ-secretase complex was elucidated by cryo-electron microscopy (cryo-EM) in 2015 [63]. However, certain regions of PSEN1 were not resolved, and the catalytic aspartates were close but not aligned properly for catalysis. Subsequently, cryo-EM structures of γ-secretase bound to APP and Notch1 substrates were elucidated [64,65], with previously visible regions of PSEN1 now resolved. Both substrates were found enveloped by PSEN1, between NTF and CTF. However, determination of these substrate-bound structures required mutation of one of the active site aspartates and cysteine mutagenesis with disulfide crosslinking between substrate and PSEN1. Thus, the enzyme was catalytically inactive, and the Cys mutations with crosslinking raised the possibility of artifacts. To trap the active enzyme without crosslinking, full TMD substrate-based mimetics were recently developed as tight-binding inhibitors of γ-secretase [62]. In the design of these structural probes, linking 10-residue helical peptide inhibitors (HPIs) to transition-state analog inhibitors (TSA) was envisioned to provide potent inhibitors that would simultaneously bind to both the docking site and active site. Another thing critical to the design was use of an HPI composed of l-amino acids to more closely mimic substrate, so the TMD mimetic would not only bind the docking site but potentially enter the interior of PSEN1 through lateral gating, as seen with substrates in the new cryo-EM studies. In this way, active enzyme would tightly bind to these TMD mimetics and trap the enzyme at the transition state, poised to carry out intramembrane proteolysis. Cryo-EM analysis of these TMD mimetics bound to γ-secretase could be carried out with proteolytically active enzyme and without the need for chemical crosslinking.

TSA **10** and HPI **11** were selected for these studies (Figure 7) [62]. TSA **10**, a pentapeptide analog spanning residues P2 through P3′, was among the most potent compounds to emerge from systematic variation of hydroxyethylurea peptidomimetics [54]. Reanalysis with purified enzyme showed **10** inhibited γ-secretase with an IC50 of 41 nM. Aib-containing HPI **11** [59], based on the TMD of APP and composed of L-amino acids, inhibited purified γ-secretase with an IC50 of 58 nM. Synthesis of the HPI–TSA conjugate with the C-terminus of **10** directly connected to the N-terminus of **11** (i.e., with no linker), did not improve the inhibitory potency. However, insertion of ω-aminoalkanoyl linkers of varying lengths improved potency and resulted in discovery of **15** (Figure 7) with an IC50 of 0.8 nM. This compound, with a 10-atom spacer, was essentially a stoichiometric inhibitor, as the concentration of γ-secretase used in the assay was 1 nM. The linker of **15** apparently contributed to potency, as the linker-**10** analog displayed an IC50 of 16 nM (cf. IC50 of **10**, 41 nM). Replacement of the hydroyxyethylurea moiety in the TSA component of **15** with a peptide bond resulted in substantial loss of potency (IC50 of 18 nM), and mass spectrometric analysis revealed that this peptide was cleaved by γ-secretase between the two Phe residues, validating the correct placement of the transition-state mimicking hydroxyethylurea moiety in the stoichiometric inhibitor **15**. Finally, disrupting the helicity of the HPI component of **15** by inverting two internal stereocenters resulted in an 8-fold loss of potency (IC50 of 6 nM). Thus, each component of **15**—the HPI, the linker, and the TSA—contributed to its high inhibitory potency.

Enzyme inhibition kinetics demonstrated that **15** had a Ki of 0.42 nM toward γ-secretase [62]. Moreover, cross-competition kinetic experiments revealed that while TSA and HPI compounds were noncompetitive with respect to each other, **15** was competitive with both TSA and HPI, consistent with interaction of **15** with both docking site and active site. This was confirmed using the biotin-tagged photoaffinity probes of TSA and HPI described earlier: TSA but not HPI prevented labeling by the TSA photoprobe, and HPI but not TSA prevented labeling by the HPI photoprobe, while **15** prevented labeling by both probes. Provocatively, conformational analysis of **15** via 2D NMR experiments showed that the HPI component is indeed in a helical conformation, while the linker and TSA components are more flexible. Among the 10 lowest energy conformations is one that overlaps well with bound APP substrate in the recently reported cryo-EM structure of γ-secretase, with the hydroxyl group of the transition-state-mimicking moiety overlapping with the scissile amide bond in the APP substrate. Taken together, these findings suggest that **15** is a TMD mimetic pre-organized for optimal binding to γ-secretase and a suitable structural probe to trap the protease complex at the transition state of intramembrane proteolysis for analysis by cryo-EM. Such studies are underway.

## 6. Functional Probes

Based on the finding that replacement of the hydroxyethylurea of **15** by a peptide bond resulted in proteolysis by γ-secretase, new helical peptides were designed based on the APP TMD as substrates for γ-secretase [66]. The *N*-terminal Aβ-like cleavage products contain Aib residues, which should induce helicity and thereby improve their solubility and detectability. Thus, these synthetic full TMD functional probes for γ-secretase were developed to allow ready analysis of all the proteolytic products by LC-MS. Peptides ranging from Gly29 to Lys55 of APP substrate (Aβ numbering) were generated, installing helix-inducing Aib residues in the *N*-terminal half of the TMD-based peptides (Figure 8). The C-terminal half of the peptides contained only APP residues, potentially allowing unwinding and binding to the active site for ε-like proteolysis followed by processive proteolysis. Initially, three such peptides (**19**–**21**) were designed, placing the Aib residues in a staggered manner in order to present different faces of the APP TMD helix to the protease.

Analysis of the proteolytic products by LC-MS revealed that ε proteolysis took place only at the two sites that occur naturally with APP: both Aβ-like peptides corresponding to Aβ48 and Aβ49 were observed along with the corresponding AICD-like peptide products [66]. Moreover, proteolytic products corresponding to Aβ43, Aβ45, and Aβ46 were also observed, suggesting that normal tripeptide trimming of the Aβ48- and Aβ49-like peptides occurred. This was validated in two ways: first, longer AICD-like products corresponding to the shorter Aβ-like products were not observed, and second, incubation of Phe-mutant versions of peptide **19** with γ-secretase demonstrated that Phe blocked cleavage wherever it was placed in the P2′ position, just as seen with APP substrate.

Because these initial functional probes were only trimmed up to the Aβ43-like site, a second round of peptides was designed in which the most C-terminal Aib in **19**, **20**, and **21** was replaced with the corresponding residue in the APP TMD (peptides **22**–**24**, Figure 8) [66]. The suspicion was that the most C-terminal Aib was preventing further trimming to Aβ40- and Aβ42-like peptides and that extending the natural APP sequence would allow another round of tripeptide trimming. Indeed, while **22**–**24** were still cleaved exclusively at the normal ε sites, generating only the expected AICD-like products, the Aβ-like products now ranged from “Aβ40” to “Aβ45”. Thus, trimming is more efficient with the second-round peptides, as no “Aβ46”, “Aβ48” or “Aβ49” were detected and trimming to “Aβ40” and “Aβ42” occurred with peptide **22**. Replacement of an additional Aib residue of **22** with the corresponding APP TMD residue led to even further trimming to “Aβ37” and “Aβ38”, as seen with APP substrate.

These helical peptide functional probes were further validated as surrogate substrates by synthesizing peptides based on **22**–**24** containing APP TMD FAD mutations V44A and I45F [66]. In APP substrate, these disease-causing mutations have opposite effects on ε cleavage site preference: V44A skews cleavage toward the Aβ48-producing site, and I45F favors the Aβ49-producing site compared to what is seen with wild-type APP. After incubation with γ-secretase, the mutant versions of **22**–**24** showed these same trends, especially the V44A and I45F mutants of **22**. Provocatively, the FAD-mutant peptides were all deficient in trimming compared to their wild-type counterparts, with longer Aβ-like peptides up to “Aβ49” being detectable. These findings are consistent with earlier studies with wild-type APP substrate and PSEN1 FAD-mutant γ-secretases, which showed deficient trimming and increased levels of long Aβ peptides [46]. Taken together, these observations raise the possibility that Aβ peptides ranging from 45 to 49 residues, which contain most of the APP TMD and are membrane-anchored, are pathogenic in FAD.

## 7. γ-Secretase Inhibitors: Therapeutic Potential

The search for γ-secretase inhibitors (GSIs) as potential therapeutics for AD had gone on for over two decades, ever since Aβ was discovered to be a normally secreted peptide produced from a variety of cell types in culture [67]. Such inhibitors were identified even before the components of the protease were known and ultimately served as critical tools for discovery of presenilin as the catalytic component [28,29], as described earlier. Initially, these inhibitors were simple peptidomimetics (e.g., TSAs), but pharmaceutical companies quickly developed compounds with much better drug-like properties that allowed in vivo testing for the ability to lower Aβ in the brains of transgenic AD mice (e.g., expressing FAD-mutant APP and presenilin).

Acute treatment with GSIs did show such proof of principle for these drug candidates [68,69]; however, chronic treatment revealed serious peripheral toxicities [70,71], such as gastrointestinal bleeding, immunosuppression, and skin lesions, all effects that could be traced to inhibition of Notch proteolysis and signaling. As AD patients would be required to take GSIs for years and perhaps decades, the severe toxic consequences of γ-secretase inhibition caused great concern. While there were hopes for a therapeutic window that would allow lowering brain Aβ levels without the peripheral Notch-deficient toxicity, the failure of one GSI, semagacestat (Figure 9), in phase III clinical trials, dashed these hopes [72]. The trial resulted in unacceptable peripheral toxicities and—more worrisome—cognition that was worse than the placebo control groups.

Because all the serious toxic effects were apparently caused by inhibition of Notch signaling, the focus then went toward finding GSIs that could selectively inhibit the proteolysis of APP by γ-secretase without affecting Notch1 proteolysis. This led to the discovery of so-called “Notch-sparing” GSIs [73,74,75,76], a misleading term, as these compounds show APP/Notch selectivity and not complete lack of effect on Notch proteolysis. Moreover, the degree of selectivity was a matter of debate, with some reports of a lack of any selectivity for APP [77,78]. One such compound, avagacestat (Figure 9), went as far as phase II clinical trials, and like semagacestat caused Notch-deficient toxicities at higher doses, equivocal Aβ lowering in cerebrospinal fluid at lower doses, and worsening of cognition [79]. Evidence from mouse models suggest that the cognitive worsening may be due to increased γ-secretase substrates [80], although elevation of total Aβ, seen in plasma at low inhibitor concentrations, may be responsible [81,82]. These findings have effectively halted further development of GSIs for AD. Interestingly though, these compounds may be repurposed for oncology, for the treatment of various cancers that involve overactive Notch signaling [83].

## 8. γ-Secretase Modulators: Therapeutic Potential

While GSIs are out of further consideration for AD therapeutics, γ-secretase modulators (GSMs) are still of keen interest [84]. These compounds (see Figure 10 for examples) have the effect of lowering Aβ42 levels without decreasing overall Aβ levels or otherwise inhibiting general γ-secretase activity [85,86]. The decrease in Aβ42 is correlated with an increase in Aβ38, thereby replacing a highly aggregation-prone form of Aβ with a much more soluble form. Thus, these compounds can prevent the formation of plaques and other higher-order assembly states of Aβ42 in the brain. GSMs, however, have no effect, even at very high concentrations, on Notch proteolysis and signaling, nor do they elevate γ-secretase substrates. Presumably for these reasons, these compounds have shown excellent safety profiles, both in animal models and in human trials.

The mechanism of action of these compounds is not entirely clear, although the correlation between Aβ42 lowering and Aβ38 elevation is relevant, as Aβ42 is a precursor to Aβ38. γ-Secretase cleaves the Aβ42 C-terminus to release a tetrapeptide [45], and isolated γ-secretase converts synthetic Aβ42 to Aβ38 with release of this tetrapeptide [87]. Moreover, presenilin mutations decrease the Aβ42-to-Aβ38 conversion while GSMs stimulate it. Thus, GSMs appear to decrease Aβ42 by enhancing the carboxypeptidase activity of γ-secretase that converts this aggregation-prone peptide to Aβ38.

A critical issue with GSMs, however, like all anti-Aβ therapeutic strategies, is the design of clinical trials [88]. So far, all reported clinical trials with candidate AD therapeutic agents, including GSIs, GSMs and anti-Aβ immunotherapy, have been with individuals who already have AD or a pre-AD condition called mild cognitive impairment (MCI). Even in those with MCI, substantial neurodegeneration has occurred, and there are serious concerns that targeting Aβ after the onset of symptoms is too late. Aβ pathology in the brain may appear more than 10 years before the clinical manifestation of AD [89]. As Aβ pathology apparently precedes tau pathology [4,90], and tau pathology may then propagate from neuron to neuron [91,92], blocking Aβ after tau pathology is initiated may not prevent or slow the progression of AD and may not even prevent or delay disease onset. For anti-Aβ strategies—including GSMs—to succeed, clearer knowledge of the pathogenic process and timing is needed, as are convenient and reliable biomarkers and diagnostics.

Moreover, the clinical success of GSMs is completely dependent on whether Aβ42 is indeed the pathogenic entity in AD. The reasons for the focus on Aβ42 are arguably more historic than a result of an objective search with no preconceptions. Over 100 years ago, Alois Alzheimer described extraneuronal amyloid plaques as a signature pathological characteristic of the disease, and the discovery in the late 1980s and early 1990s that the primary protein component of the plaques is Aβ [93,94], with Aβ42 being the predominant species [13] and most aggregation-prone [95], led to the assumption that Aβ42 is the likely pathogenic species. Subsequent studies ranging from effects of APP and presenilin mutations to the neurotoxicity of various Aβ assemblies would seem to confirm this hypothesis. However, selectivity in the reporting of findings (negative results are more difficult to publish) in combination with incomplete knowledge of all forms of Aβ could result in mistaking correlation for causality. In light of more recent findings, mentioned earlier, that FAD mutations in PSEN1 increase Aβ peptide intermediates of 45 residues and longer, it would seem worthwhile to determine the effects of GSMs on the first or second trimming events of APP substrate by γ-secretase (e.g., Aβ49→46, Aβ45→42).

## 9. Summary and Perspective

The devastation of AD is severe, the numbers afflicted are large, and the need for effective therapeutics is great. Despite decades of intense efforts by laboratories around the world, in academia, government, and industry, only symptomatic treatments for AD are available, and no drugs of any kind have been approved since 2003. The large majority of clinical trials have involved agents that target Aβ in some way, to block its production or aggregation or to stimulate its clearance. γ-Secretase remains a top target, especially as altered processing of APP substrate by this protease complex is clearly involved in the pathogenesis of FAD. Dominant missense mutations in presenilins—the catalytic component of the γ-secretase complex—and APP—the γ-secretase substrate precursor to Aβ—cause FAD with virtually 100% penetrance. A single mutant allele in either the enzyme or substrate that produce Aβ fates the carrier to AD in midlife. The pathology, presentation, and progression of this genetic form of the disease is essentially indistinguishable from the much more common form that strikes in late life. Thus, it seems likely that Aβ in some form similarly plays a key role in the etiology of sporadic late-onset AD.

Given the close similarities between the familial and sporadic forms of the disease, elucidating the pathogenic mechanisms of the former—a simpler problem—is likely to be illuminating for the latter. Early work had focused on the aggregation-prone Aβ42, as this Aβ variant is the principal component of the characteristic cerebral plaques of AD. In many cases, FAD mutations can elevate the ratio of Aβ42 to the more soluble Aβ40. However, processing of APP substrate by γ-secretase is complex, involving carboxypeptidase trimming of initially formed Aβ48 or Aβ49, and FAD mutations can elevate longer forms of Aβ that contain most of the APP TMD and remain membrane-bound. A critical question is whether these long Aβ peptides play an important role in the pathogenic process.

A variety of substrate-based probes for γ-secretase have been developed. Transition-state analog inhibitors (TSAs) provided the first clue that the enzyme is an aspartyl protease, and affinity labeling reagents based on TSAs covalently bind to presenilin NTF and CTF, early evidence that the active site resides between these two subunits. TSAs also helped characterize active site pockets and suggested the existence of a separate initial substrate docking exosite. Helical peptide inhibitors (HPIs) designed to interact with this docking site identified the presenilin NTF/CTF interface as its location and close proximity to the active site. More recently, linking HPI to TSA has provided full substrate TMD-based stoichiometric inhibitors as structural probes for cryo-EM analysis to trap the protease complex in its transition state for intramembrane proteolysis. Full substrate TMD-based functional probes have also been developed to facilitate analysis of all proteolytic products generated during the complex processing of APP substrate by γ-secretase. Toward therapeutics targeting γ-secretase, a wide variety of small-molecule inhibitors (GSIs) and modulators (GSMs) have been reported, with some going through clinical trials. GSIs can effectively lower Aβ production in vivo. However, GSIs also interfere with critical Notch signaling, causing severe side effects. More concerning, GSIs cause cognitive worsening. These devastating failures of GSIs in late-stage clinical trials has turned the field toward GSMs. Modulation rather than inhibition appears to be safe and allows specific targeting of Aβ42. These compounds apparently stimulate the Aβ42→38 trimming step, to lower the aggregation-prone peptide. Whether GSMs will prevent AD remains to be determined, however. If they are efficacious, this would be a strong argument for Aβ42 being the pathogenic entity. If they are not, exploring the effects of GSMs on longer forms of Aβ and testing potential pathogenic roles for these membrane-associated forms of the peptide would seem to be worthwhile.

## Figures and Tables

**Figure 1 molecules-26-00388-f001:**
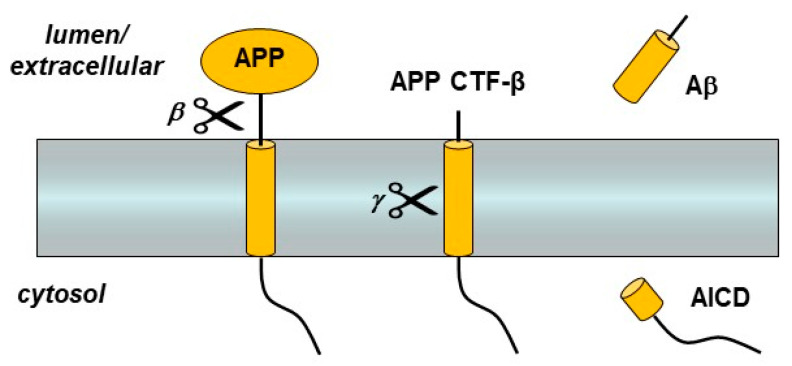
Amyloid precursor protein (APP) processing by β- and γ-secretases. The single-pass membrane protein APP is proteolyzed just outside the transmembrane domain (TMD) by β-secretase. The remaining membrane-bound C-terminal fragment (APP CTF-β) is then cleaved within the TMD to produce the amyloid β-peptide (Aβ) and the APP intracellular domain (AICD).

**Figure 2 molecules-26-00388-f002:**
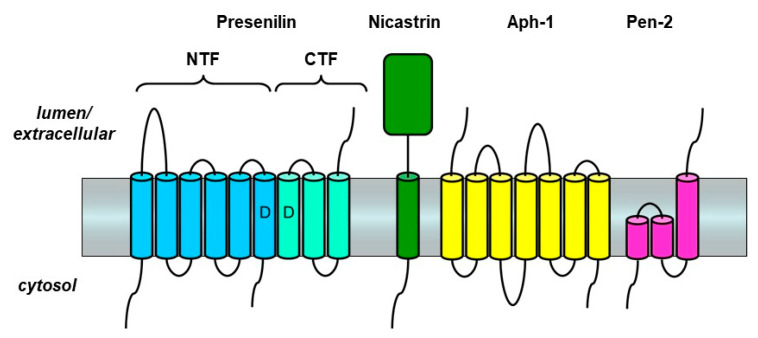
Presenilin and other components of the γ-secretase complex. Presenilin is a 9-TMD protein that contains two TMD aspartates (D) essential to catalysis. Assembly of presenilin with the three other components—nicastrin, Aph-1, and Pen-2—triggers autoproteolysis of presenilin into an N-terminal fragment (NTF) and C-terminal fragment (CTF) to form the active γ-secretase complex.

**Figure 3 molecules-26-00388-f003:**
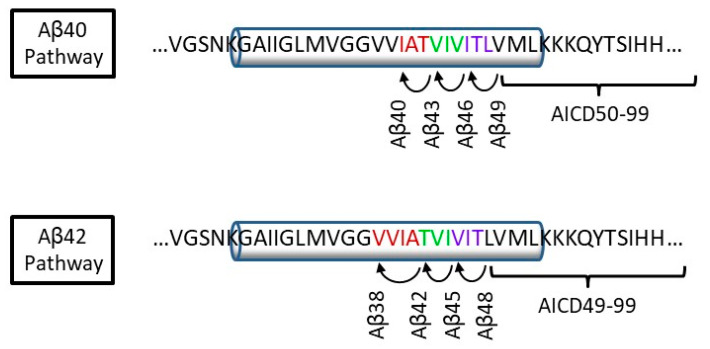
Processive proteolysis of APP substrate by γ-secretase. Initial endoproteolysis at the ε cleavage site of APP substrate C99 results in Aβ48 or Aβ49 and corresponding APP intracellular domain fragments AICD49-99 and AICD50-99. The carboxypeptidase activity of γ-secretase then trims the initial Aβ peptides along two pathways: Aβ49→Aβ46→Aβ43→Aβ40 and Aβ48→Aβ45→Aβ42→Aβ38.

**Figure 4 molecules-26-00388-f004:**
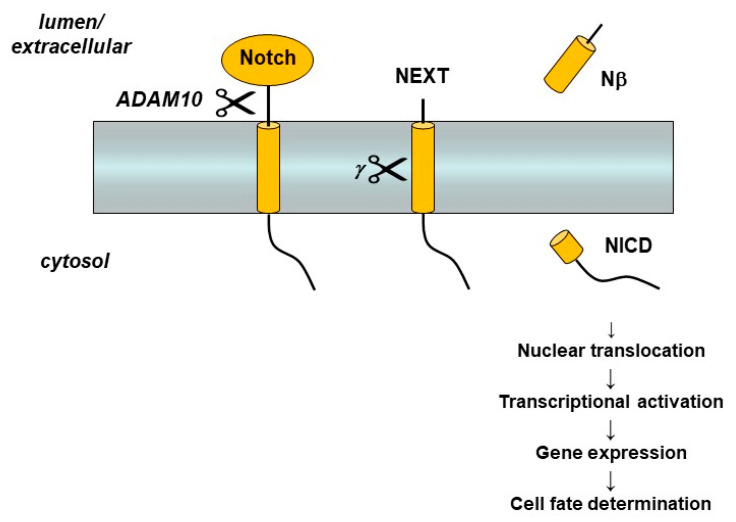
Notch receptor processing and signaling. Interaction with a cognate ligand on a neighboring cell triggers Notch ectodomain shedding by the metalloprotease ADAM-10 and then cleavage of the Notch extracellular truncation (NEXT) fragment with its single TMD by γ-secretase. Release of the Notch intracellular domain (NICD) leads to translocation to the nucleus, activation of transcriptor factors, and gene expression that controls cell differentiation.

**Figure 5 molecules-26-00388-f005:**
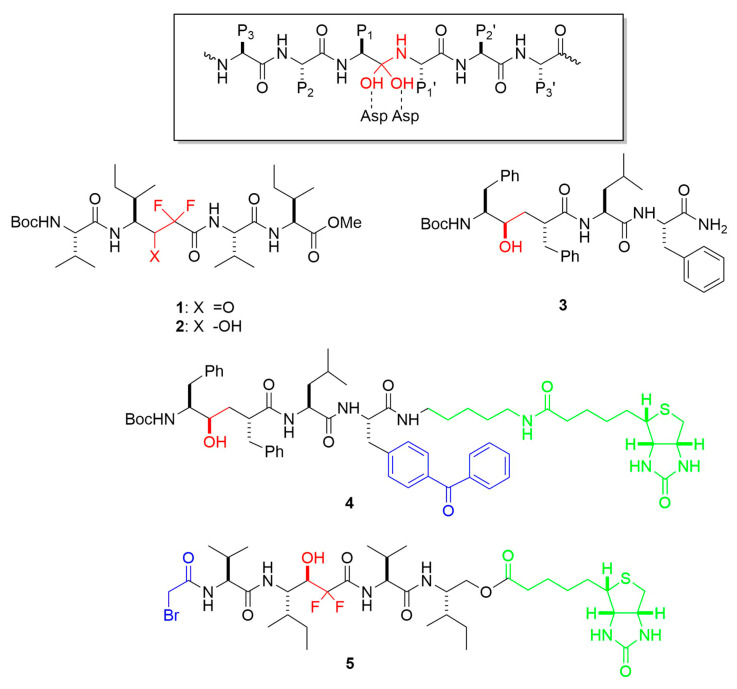
Representative transition-state analog inhibitors (TSAs) and affinity labeling reagents for γ-secretase. Functional groups that mimic the transition state of aspartyl protease catalysis are highlighted in red. Reactive groups leading to covalent attachment are blue, and linker-biotin moieties are in green. Boxed is the general structure of the tetrahedral intermediate formed upon addition of water to the scissile amide bond during aspartyl protease catalysis. Residues P1, P2, P3 and P1′, P2′, P3′ relative to the scissile amide bond are indicated.

**Figure 6 molecules-26-00388-f006:**
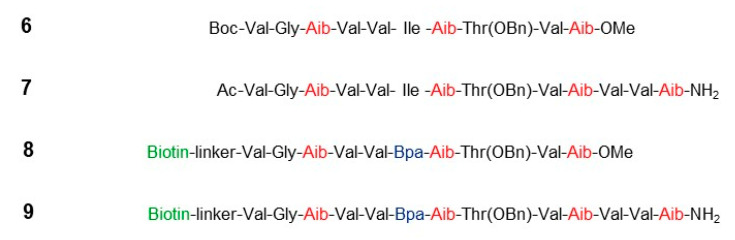
Representative helical peptide inhibitors (HPIs) and affinity labeling reagents for γ-secretase. Each depicted HPI is a d-peptide. Helix-inducing aminoisobutyric acid (Aib) residues are highlighted in red. Photoreactive 4-benzoyl-d-phenylalanine residues are blue, and linker-biotin moieties are in green.

**Figure 7 molecules-26-00388-f007:**
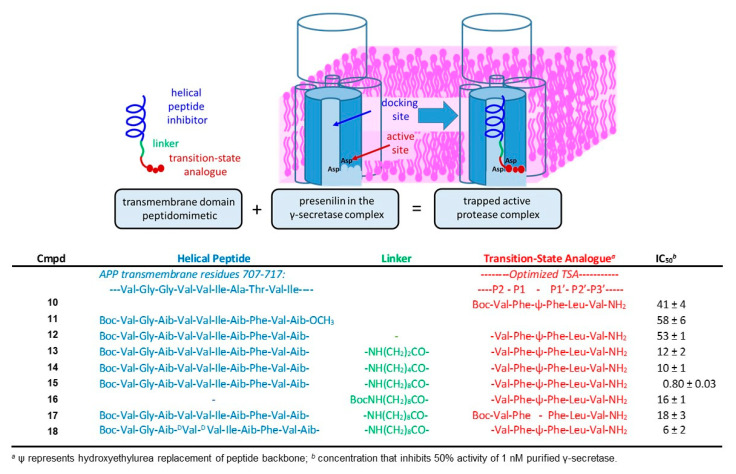
Discovery of structural probes for γ-secretase. Schematic shows the design concept. Helical peptide inhibitors (HPIs) directed to the substrate docking exosite were conjugated through a variable linker to transition-state analogue inhibitors (TSAs) directed to the active site. Presenilin (blue-grey) and other components of the γ-secretase complex (outlined) are shown schematically in the absence and presence of a hybrid HPI-TSA inhibitor. The active site contains two catalytic transmembrane aspartates and three hydrophobic pockets S1′, S2′ and S3′ that must be engaged for intramembrane proteolysis. Linking TSA **10** to HPI **11** through a 10-atom linker led to stoichiometric inhibitor **15**.

**Figure 8 molecules-26-00388-f008:**
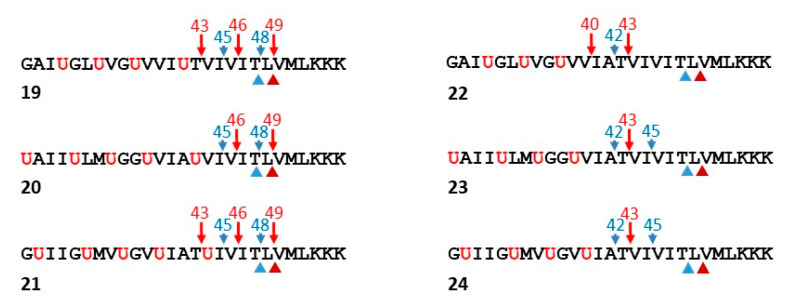
Functional probes for γ-secretase. Peptides based on the TMD of APP were designed with helix-inducing α-aminoisobutyric acid (Aib) installed in the N-terminal region. Proteolytic products were detected by LC-MS. Endoproteolytic (ε) sites, indicated by the arrowheads below the peptide sequence, were determined by analysis of AICD-like products. Trimming sites, indicated by the arrows above the peptide sequence, were determined by analysis of Aβ-like products. Note that replacement of the most C-terminal Aib in peptides **19**–**21** with the corresponding APP TMD residue (peptides **22**–**24**) leads to most effective trimming, producing shorter Aβ-like products.

**Figure 9 molecules-26-00388-f009:**
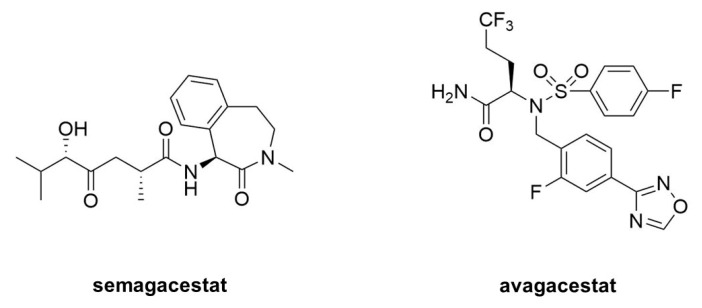
Clinical candidate γ-secretase inhibitors (GSIs). Semagacestat (**left**) is nonselective for APP vis-à-vis Notch, while avagacestat (**right**) is reported as selective for blocking γ-secretase proteolysis of APP over Notch.

**Figure 10 molecules-26-00388-f010:**
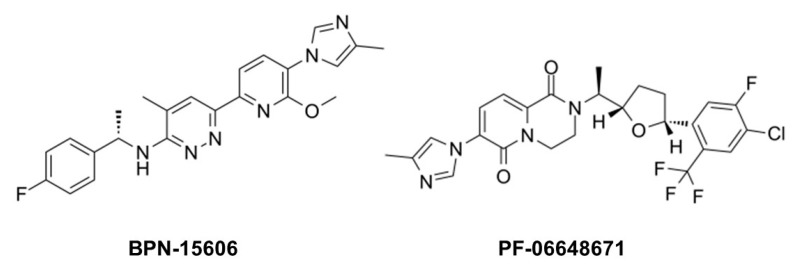
Example γ-secretase modulators (GSMs).

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
