# Peer review of "Probing Mechanisms and Therapeutic Potential of γ-Secretase in Alzheimer’s Disease"

_molecules, 2021, doi:10.3390/molecules26020388_

Round 1

Reviewer 1 Report

In this review Dr Wolfe summarizes twenty years of research on gamma-secretase and highlights the work from his own laboratory, in particular ongoing efforts to pin down the mechanism of this proteolytic complex using inhibitors and substrate probes. The paper is very well written and illustrated. However, most references are not included in the text and a few of those inserted are listed according to author names whereas the literature section is listed by numbers.The ref citations in the text start at line 120 and ends at line 141. Numbers then appear at line 382!

A few other minor points may be addressed.

  1. The abstract states that gamma-secretase processes nearly 100 substrates. Güner and Lichtenthaler have recently recensed 149 gamma-substrates.Semin Cell Dev Biol. 2020 Sep;105:27-42. doi: 10.1016/j.semcdb.2020.05.019.
  2. The text at line 121 refers to a first APP cleavage at position epsilon. For consistency, it would be helpful to indicate this epsilon site on Figure 3 (or at least to mention it in the legend after "Initial endoproteolysis").
  3. The section 4 - Chemical probes - details the binding of the substrate to the gamma-secretase active site. To facilitate the reader's understanding, Figure 5 could include an extended diagram of the APP substrate cleavage region to indicate the P1, P2, P3, P'1, P'2 and P'3 residues.

Author Response

Reviewer 1

In this review Dr Wolfe summarizes twenty years of research on gamma-secretase and highlights the work from his own laboratory, in particular ongoing efforts to pin down the mechanism of this proteolytic complex using inhibitors and substrate probes. The paper is very well written and illustrated. However, most references are not included in the text and a few of those inserted are listed according to author names whereas the literature section is listed by numbers.The ref citations in the text start at line 120 and ends at line 141. Numbers then appear at line 382!

Response: Please accept my apologies.  I had mistakenly uploaded a penultimate draft that lacked complete references and was not properly formatted.  I was alerted to this by the editors at Molecules and had immediately sent them the final version with completed references and properly formatted, but this corrected version had apparently not been uploaded in time for this reviewer.  The final version has 95 references properly formatted.

A few other minor points may be addressed.

  1. The abstract states that gamma-secretase processes nearly 100 substrates. Güner and Lichtenthaler have recently recensed 149 gamma-substrates.Semin Cell Dev Biol. 2020 Sep;105:27-42. doi: 10.1016/j.semcdb.2020.05.019.

Response: This recent review is cited (reference 55), and I have modified this sentence to say “nearly 150 substrates”, both in the abstract on line 7 and in the main text on line 183.

  1. The text at line 121 refers to a first APP cleavage at position epsilon. For consistency, it would be helpful to indicate this epsilon site on Figure 3 (or at least to mention it in the legend after "Initial endoproteolysis").

Response: The legend in Figure 3 (line 133) has been so modified.

  1. The section 4 - Chemical probes - details the binding of the substrate to the gamma-secretase active site. To facilitate the reader's understanding, Figure 5 could include an extended diagram of the APP substrate cleavage region to indicate the P1, P2, P3, P'1, P'2 and P'3 residues.

Response: Figure 5 has been so modified.  The corresponding part of the legend (lines 201-202) has also been modified to reflect this change.

Reviewer 2 Report

The present account (“Probing Mechanisms and Therapeutic Potential of γ-Secretase in Alzheimer’s Disease”), submitted by M. S. Wolfe, a well known and active scientist in the area, is certainly an interesting review describing the chemical probes and therapeutic prototypes based on γ-secretase for the potential therapy of AD. Thus, I suggest to accept the manuscript for publication, but after major revision.

Personally I don’t believe in Ab or tau hypothesis as the key and central hypotheses for AD, as no single drug has been approved till now for AD therapy based on them. Last hopes on “aducanumab” are frustrated...

Something is wrong, and it is time move forward or refresh old therapies.

I’m convinced that if the past, present and enormous efforts dedicated to the Ab, would have been invested in finding non-toxic tacrine-like ligands, the situation of the therapy for AD would been totally different…Thus, the point is to improve what works: to recover the level of neurotransmitters by inhibiting the ChEs, modulating the oxidative stress…Yes, I know this is old-fashioned…, but unfortunately  the only strategy that has afforded drugs for AD patients.

Thus, as it is obvious that Aβ42 is not the only pathogenic entity in AD, a more (auto)-critical analysis would be appreciated, necessary and of interest to have a more useful review.

Author Response

Response: This reviewer apparently did not read the manuscript carefully.  I specifically called into question the pathogenicity of Aβ42, raising the possibility that little-studied membrane-anchored forms of Aβ of 45 to 49 amino acids in length may be pathogenic, as evidence is emerging that FAD mutations lead to deficient carboxypeptidase trimming of these long Aβ peptides.  Please see the last paragraphs in section 6 (on functional probes) lines 385-391, section 8 (on γ-secretase modulators) lines 466-480, and section 9 (summary and perspective) lines 497-505, 522-529.  Please also refer to the summary and perspective section in which I lay out compelling evidence that some form of Aβ is pathogenic in familial Alzheimer’s disease, as this hereditary form of the disease is caused by missense mutations in the substrate and enzyme that produce Aβ, and that therefore some form of Aβ is also likely to be pathogenic in sporadic late-onset Alzheimer’s disease (lines 490-496).  The reviewer is correct that “something is wrong”.  I raise the possibility here that this “something” may be ignoring these long, membrane-anchored Aβ peptide intermediates.

Round 2

Reviewer 1 Report

The author has corrected the issue with the references and has addressed every point I had raised. I am satisfied with this current revised version.

Reviewer 2 Report

This autor has not conveniently addressed this reviewer suggestions. Please, reject, and publish elsewhere.